# Sustainability of Vehicle Fuel Biomethane Produced from Grass Silage in Finland

**Saija Rasi [1],\*, Karetta Timonen [2], Katri Joensuu [2], Kristiina Regina [3], Perttu Virkajärvi [4], Hannele Heusala [2], Elina Tampio [2]** 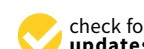 **and Sari Luostarinen [3]**

1   Natural Resources Institute Finland (Luke), Production Systems, Survontie 9 A, FI-40500 Jyväskylä, Finland
2   Natural Resources Institute Finland (Luke), Production Systems, Maarintie 6, FI-02150 Espoo, Finland; karetta.timonen@luke.fi (K.T.); katri.joensuu@luke.fi (K.J.); hannele.heusala@luke.fi (H.H.); elina.tampio@luke.fi (E.T.)
3   Natural Resources Institute Finland (Luke), Bioeconomy and Environment, Tietotie 4, FI-31600 Jokioinen, Finland; kristiina.regina@luke.fi (K.R.); sari.luostarinen@luke.fi (S.L.)
4   Natural Resources Institute Finland (Luke), Production Systems, Halolantie 31 A, FI-71750 Maaninka, Finland; perttu.virkajarvi@luke.fi
\*   Correspondence: saija.rasi@luke.fi; Tel.: +358-29532-6469

**Abstract:** Increasing demand of fossil-free fuels in the transport sector drives towards using new biomass sources in fuel production. Municipal waste as a substrate is used in many countries in biomethane production, but the amount of waste can cover only a small portion of the fuel used. In Europe, the new renewable energy directive (RED II) was established December 2018 to ensure the sustainability of renewable fuels. The directive includes typical and default greenhouse gas (GHG) emissions for several potential substrates, such as biogas from manure or maize silage, which the biogas plants can use to verify their emissions directly or to calculate their emissions using the methods provided. However, such default value for grass silage as biogas substrate is lacking. We defined the conditions needed to fulfil the sustainability criteria of the directive when producing biomethane for vehicle fuel using grass silage as the feedstock in Finland. The emission reduction targets are not easy to achieve in Finland when using grass cultivated exclusively for energy production. The reduction targets can be achieved, however, if the grass is cultivated due to an improved crop rotation, where the grass is co-digested with manure and/or energy sources with zero emissions for the process can be applied.

**Keywords:** biogas; biomethane; grass silage; manure; RED II directive; sustainability; vehicle fuel

## 1. Introduction

Biomethane is a renewable alternative to natural gas to be used, for example, as a vehicle fuel replacing fossil fuels. It can be produced from various organic materials via anaerobic digestion (AD). Biomethane is thus upgraded biogas from which carbon dioxide ($CO_2$) and possible contaminants are removed to reach a methane content of over 95% [1]. Biogas production from municipal waste materials and manure is common practise in many countries and, for example, maize silage is broadly used as a substrate in German biogas plants, of which 95% use agricultural substrates [2]. The most common biogas use is the production of electricity or combined heat and electricity (CHP), but upgrading to biomethane is increasing, for example, in Germany and Sweden [3].

The European Union new renewable energy directive (RED II) [4] for the years 2021–2030 was released in December 2018. The directive sets the overall EU target for the consumption of Renewable Energy Sources by 2030 to 32%. It also sets a sub-target of a minimum renewable energy proportion of

14% of the energy consumed in road and rail transport by 2030. Finland is committed to reducing transport emission by 50% by 2030 [5]. For that, both electric vehicles and biofuels are needed. Finland has also set the target of 50,000 gas driven passenger cars by 2030 [6].

To produce the needed biomethane, the Finnish biogas sector needs development. Separately collected municipal biowaste and sewage sludge are already mainly used in biogas production [7], and even if all biogas produced from them was upgraded to biomethane, new substrates are needed to fill the demand for low emission vehicles in the future. Manure has potential as a substrate for AD, and the treatment of manure in AD also improves the manure management through, e.g., enhanced fertiliser value and decreased risk of nitrate leaching to water bodies as organic nitrogen is transformed into ammonium nitrogen ($NH_4$-N). The relatively low methane potential in manure [8], however, can create economic barriers to the biogas plants, and often co-substrates are needed to increase the energy production and thus the profitability. These co-substrates could also be of agricultural origin.

Although it is a common substrate in Europe, maize is not an optimal energy crop in Finland due to climatic constraints. In countries with minor maize production (e.g., Ireland, Denmark Sweden, Finland), grass silage is seen as a more optimal option for gas production. Perennial grass swards fit well to the Finnish growing conditions, as the grasses start growing early in the spring when solar radiation is abundant and soil water situation is good. In addition to animal feed production, grasses are grown as perennial green fallows and in buffer zones. Silage production is closely connected to milk production and spare grass silage, and potential additional biomass generated from more intensified grass production could be made available on cattle production areas. Green fallows are located quite evenly around the country in relation to overall field area [9].

The sustainability of biogas production is widely studied, and especially with manure and waste materials as substrates, the emissions in fuel use can be expected to decrease significantly compared to using fossil fuels [10,11]. However, the situation becomes more complicated when biomass is cultivated exclusively for energy production. In some cases, land use changes related to replacing land area from food and feed production to energy production purposes have been seen as problematic. Additionally, emissions from plant cultivation can increase the overall emissions of the fuel produced and used.

The RED II defines a series of sustainability and GHG emission criteria that the fuels used in transport must comply with to be included into the overall 14% target and to be eligible for financial support from public authorities. According to the directive, biomethane used as a vehicle fuel is renewable (and can be counted in the overall target) if the emission reduction compared to fossil fuels is over 65%. This applies to biogas plants in operation in 2021 or later. The directive includes typical and default GHG emissions for several potential substrates, such as biogas from manure or maize silage, which the biogas plants can use to verify their emissions directly or to calculate their emissions using the methods provided [4]. However, such default information for grass silage is missing.

The aim of this work was to assist northern European countries fulfilling the requirements of the RED II directive considering grass silage as a potential substrate for biomethane production. We estimated the GHG emission reductions of different types of biomethane production scenarios with comparison to fossil fuels as defined by the directive to facilitate selection of the most sustainable production chains for future biofuel production. The emission reduction scenarios for different production chains of biomethane from grass silage, produced in short term ley farming system, assist, for example, biogas plant operators to select the most favourable substrates and operational practices to ensure sustainable bioenergy production.

## 2. Materials and Methods

Life cycle assessment (LCA) methodology was used to calculate the GHG emissions and the climate impact of grass ley cultivation, harvest, preservation for silage, and biomethane production. The calculation was done according to RED II [4] and IPCC [12] instructions and International Standards for LCA (ISO 2006a, ISO 2006b) and by applying system constraints and other assumptions defined below. The assessed impact category of climate impact included carbon dioxide ($CO_2$), methane ($CH_4$),

and dinitrogen monoxide ($N_2O$) emissions. The results of the life cycle inventory were characterised using factors suggested in RED II [4], i.e., $CO_2 = 1$, $CH_4 = 25$ and $N_2O = 298$.

## 2.1. System Description and Boundaries

The climate impact of AD using grass silage or mixture of grass silage and manure as substrates was analysed for the entire life cycle using the functional unit of $MJ_{energy}$. The production methods of grass included grass (a mixture of timothy (*Phleum pratense*) and meadow fescue (*Festuca pratensis*)) and mixtures of grass and clover (red clover (*Trifolium pratense*)) as well as grass harvested from buffer zones or field for green manuring. The biomass is later referred to as grass or silage irrespective of its botanical composition.

The emissions from grass cultivation to biomethane production and use were included, as they are cultivation, transportation of grass silage from fields to biogas plant, digestion process, and upgrading and compression of biogas to produce biomethane for fuel (Figure 1). According to RED II, biogas plants using manure as a co-substrate can include negative emissions for emissions saved from raw manure management ($-45$ g$CO_{2eq}$/MJ). This was included in the calculations (expressed as a dashed line in Figure 1). The use of digestate as a fertiliser was not considered, except in cases of green manuring.

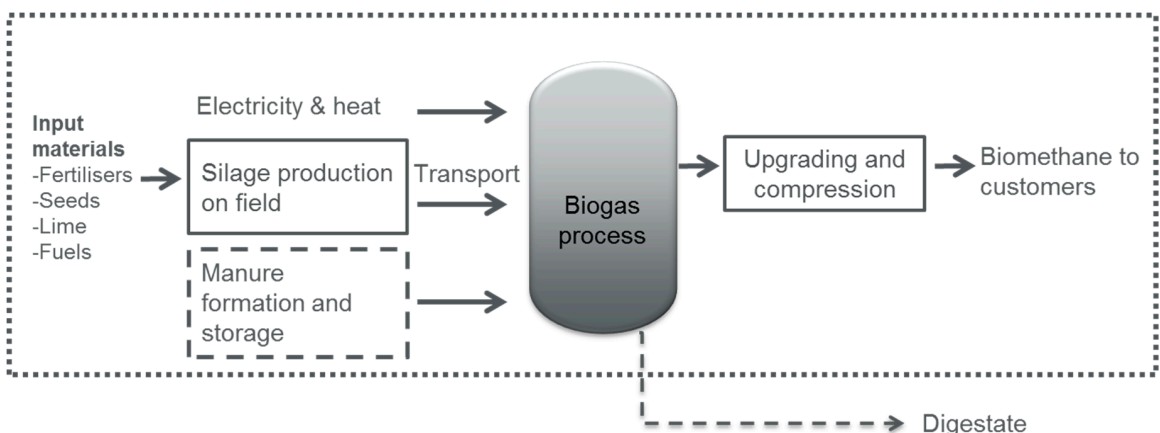

**Figure 1.** System boundary of the study. Of the different scenarios considered, only with green manuring is the digestate transported back to fields.

Four different substrate compositions for digestion (scenarios) were compared (Table 1). The scenarios were selected to present alternative cases with grass silage as a substrate for biogas production. All scenarios were based on agricultural materials only. In Finland, grass is cultivated both in mineral and organic soil types, and their GHG emissions differ significantly. Therefore, separate scenarios (1, 2, 6, and 7) for these soil types were assessed, although, in practice, the soil types can vary within an individual farm. For clover silage (3 and 8), only mineral soil was included due to clover not thriving on organic soils. Green manuring (5) was also included to represent a situation where the farm cultivates grass to improve crop rotation and thus maintenance of good soil structure. Grass from green manuring is often ploughed and incorporated into the soil, but it could also be harvested for biogas use. Manure as a co-substrate was included to illustrate the effect on GHG emissions when the share of manure in feed is high (4) or low (6–8).

The emissions from the various phases of the biomethane production chain and the total emissions from the entire chain were allocated to the end-product, i.e., MJ of transport fuel produced. The emission reduction of the production chain is calculated by comparing its total emissions with the fossil fuel comparator value for transport fuel (94 g$CO_2$eq/MJ) given in RED II [4] (Equation (1)).

$$Emission\ reduction = \frac{fossil\ fuel\ comparator - biomethane\ total\ emissions}{fossil\ fuel\ comparator} \tag{1}$$

**Table 1.** Scenarios for climate impact assessment in biomethane production from grass silage (and manure) under northern conditions.

| Scenario | Amount of Substrates (t/a) | Energy (MWh) |
|---|---|---|
| 1. Grass silage (mineral soil) | 62,000 | 46,100 |
| 2. Grass silage (organic soil) | 62,000 | 46,100 |
| 3. Clover silage (mineral soil) | 74,000 | 45,900 |
| 4. Grass silage (mineral soil) + 80% manure * | 135,000 | 46,670 |
| 5. Grass silage from green manuring | 48,000 | 46,000 |
| 6. Grass silage (mineral soil)+ 20% cattle manure | 74,000 | 46,000 |
| 7. Grass silage (organic soil)+ 20% cattle manure | 74,000 | 46,000 |
| 8. Clover silage + 20% cattle manure | 87,000 | 45,550 |

* cattle manure 20%, solid cattle manure 20%, pig slurry 40%.

## 2.2. Grass Cultivation

Inventory data for the field production of grass silage were obtained from Finnish FootprintBeef-project that based the cultivation data on a database collected from Finnish cattle farms during 2002–2011 by Pro Agria Advisory Centres and included details on cultivation measures and soil types at the field parcel level. For grass and clover, the yield was assumed to be similar to the boundary value of the highest yield quartile in the farm data (Table 2). For green manuring, the boundary value of the lowest yield quartile was used. For fertilisation levels, Finnish average values were used. Because of its symbiotic nitrogen fixation, the nitrogen fertilisation of clover leys can be reduced to 50% of that of grass leys. The annual lime use was 101 kg/ha of which 76% was applied as limestone, 21% as dolomite lime stone, and 2% as suitable side streams. Use of field machinery was estimated based on number of harvests per year (Table 2) and the length of the crop rotation, which determined the frequency of other field operations. For those other than grass from the buffer zone, it was assumed that the grass was established with a cereal as a cover crop in the first year, followed by three years of grass production for silage and finally ploughing the field for a subsequent cereal. The use of AIV preservative (formic acid buffered with ammonium formate) for the harvested grass was estimated to be 5 L/kgTS$_{silage}$ [13]. The GHG emission factors for input materials and fuels are presented in Table 3. For diesel fuel production, the emission factor was obtained from the Ecoinvent 3 database [14]. In the scenario with green manure grass used as the substrate for biogas production, only the yield level and the emissions caused by harvesting were taken into account, as grass is seen as a side product from green manuring.

**Table 2.** Annual total solids (TS) yields per ha, nitrogen fertilisation, and number of harvests of different production systems assessed in this study.

| Yield Level | Mineral Soil | | Organic Soil | Green Manuring |
|---|---|---|---|---|
| | Grass ley | Clover ley | Grass ley | |
| Yield level (kgTS/ha) | 7530 | 7530 | 7530 | 3040 |
| Nitrogen fertilisation (kgN/ha) | 180 | 90 | 130 | |
| Number of harvests per year | 3 | 3 | 3 | 1 |

**Table 3.** Emission factors and sources for the production of inputs and emissions during cultivation.

| Emission Category | Emission Factors | Reference |
|---|---|---|
| **Production of input materials and fuels** | | |
| Mineral fertiliser | 3.6 kg $CO_{2eq}$/kg N | [15] |
| Limestone | 0.01 kg $CO_{2eq}$/kg | [16] |
| Seeds | 0.5 kg $CO_{2eq}$/kg | [17] |
| AIV preservative | 3.1 kg $CO_{2eq}$/kg | [18] |

**Table 3.** *Cont.*

| Emission Category | Emission Factors | Reference |
|---|---|---|
| **Emissions during field production** | | |
| Direct $N_2O$ from fertilisation | N input in fertilisers (kg N/ha)x0.01x(44/28) $kgN_2O$/ha | [12] |
| Indirect $N_2O$ from N leaching | N input in fertilisers (kg N/ha)x0.0075x0.3x(44/28) $kgN_2O$/ha | [12] |
| Indirect $N_2O$ from N volatilisation as $NH_3$ and $NO_x$ | N input in fertilisers (kg N/ha)x0.1x0.01x(44/28) $kgN_2O$/ha | [12] |
| $N_2O$ from decomposition of crop residues | N input in crop residues (kg N/ha)x0.01x(44/28) $kgN_2O$/ha | [12] |
| $N_2O$ from decomposition of organic matter | 9.5 kg/ha x(44/28) $kgN_2O$/ha (for perennial crops) | [12] |
| Liming | 0.1 kg $CO_2$-C/ha x(44/12) | [12] |

### 2.3. Transportation and Machinery

Estimates of fuel consumption in machinery work operation were based on Mikkola and Ahokas [19] for ploughing, spreading of fertilisers, and harvesting, and Grönroos and Voutilainen [18] for spreading of lime. The direct emissions caused by machinery work and transportation were estimated based on LIPASTO database [20]. Transport during cultivation operations was included in the emission from cultivation. Separate transport emissions presented in the results section included loading the substrates on a semi-trailer truck (25 t) and transporting them from the farm to the biogas plant (20 km). The semi-trailer truck was expected to return with an empty load, as the digestate was not considered in the scenarios (except for green manuring, case 5). It was also assumed that the silage (and manure) was loaded into a biogas reactor (fuel consumption 0.06 L/m3), where the biogas formed was fed to a gas storage connected to a post-digestion tank and further to a transport fuel processing unit based on water scrubbing technology. All considered manure types were also mixed before loading, and this was estimated to have a fuel consumption of 0.3 L/m$^3$.

### 2.4. Energy and Mass Balance of the Biogas Production Chain

The amount of energy produced by AD plant was calculated using the methane production potential and the organic matter (VS) content of the substrates (Table 4). The methane yield received was multiplied by a factor of 90% so that the yield corresponded to the most likely methane yield in a real biogas plant considering their usual efficiency.

**Table 4.** The average characteristics; TS, VS, biomethane potential (BMP), and nutrient content (total nitrogen, ammonium nitrogen, and phosphorus) of the substrates according to experimental data of Natural Resources Institute Finland.

| | TS (%) | VS (%) | $N_{tot}$ (g/kg$_{ww}$) | $NH_4$-N (g/kg$_{ww}$) | $P_{tot}$ (g/kg$_{ww}$) | BMP m$^3$CH$_4$/tVS |
|---|---|---|---|---|---|---|
| Grass silage | 30.0 | 27.0 | 7.7 | 0.3 | 0.87 | 310 |
| Clover silage | 26.2 | 24.1 | 11.0 | 0.4 | 0.69 | 290 |
| Grass silage from green manuring | 40.0 | 36.0 | 5.0 | 0.4 | 0.64 | 300 |
| Cattle slurry | 9.0 | 7.2 | 5.0 | 2.9 | 0.9 | 210 |
| Solid cattle manure | 30.1 | 25.6 | 5.4 | 1.9 | 1.0 | 200 |
| Pig slurry | 8.2 | 7.0 | 4.6 | 2.9 | 1.0 | 320 |

The energy consumption of the biogas plant was assessed by estimating the energy needed to heat the substrate (from 12 to 40 °C, Equation (2)) without hygienisation. The potential use of heat exchangers was not taken into account.

$$\Delta E = c \times m \times \Delta t, \tag{2}$$

where

$\Delta E$ = energy needed for heating

$c$ = specific heat capacity kJ/kg°C ($c_{water}$ = 4.18 kJ/kg°C)

$m$ = mass, kg

$\Delta t$ = temperature change, °C

Electricity consumption for the pre-treatment was assumed to be 150 kWh/tTS, and for the biogas reactor, 3% of the energy produced by the plant [21]. The energy used for biogas upgrading was 0.70 kWh per cubic meter of methane produced, which also includes gas pressurisation at the filling station. It was assumed that biomethane was sold next to plant, thus there were no emissions from biomethane transportation. All the biogas produced was assumed to be used as a vehicle fuel, and therefore the energy needed in the biogas plant and gas upgrading and compression was calculated based on average Finnish electricity consumption [22]; specific energy emission factors were obtained from Official Statistics Finland [23] and Ecoinvent database [14]. Emissions of heat consumption were from combustion of wood chips for energy (6 g$CO_{2eq}$/MJ) [4].

## 3. Results and Discussion

The scenarios with the highest total GHG emissions (Figure 2) were the ones when grass silage was produced from organic soils (scenarios 2 and 7, 104 and 100 g$CO_{2eq}$/MJ, respectively). This was due to $N_2O$ emissions caused by the decomposition of soil organic matter accounting for 70% of the emissions at the cultivation phase (Table 5). For grass and clover silage from mineral soils (scenarios 1 and 3), the majority of the emissions (45 and 34 g$CO_{2eq}$/MJ, respectively) were caused by $N_2O$ emissions from nitrogen fertilisation (Table 5). These total emissions are comparable to the emissions of maize-based biogas reported by Adams and McMagnus [24] (26.8–43.9 g$CO_{2eq}$/MJ depending on the scenario) despite some calculation principles, such as methane losses, being different. In the scenario with minimum methane losses, the total emissions for maize were 30.2 g$CO_{2eq}$/MJ [24].

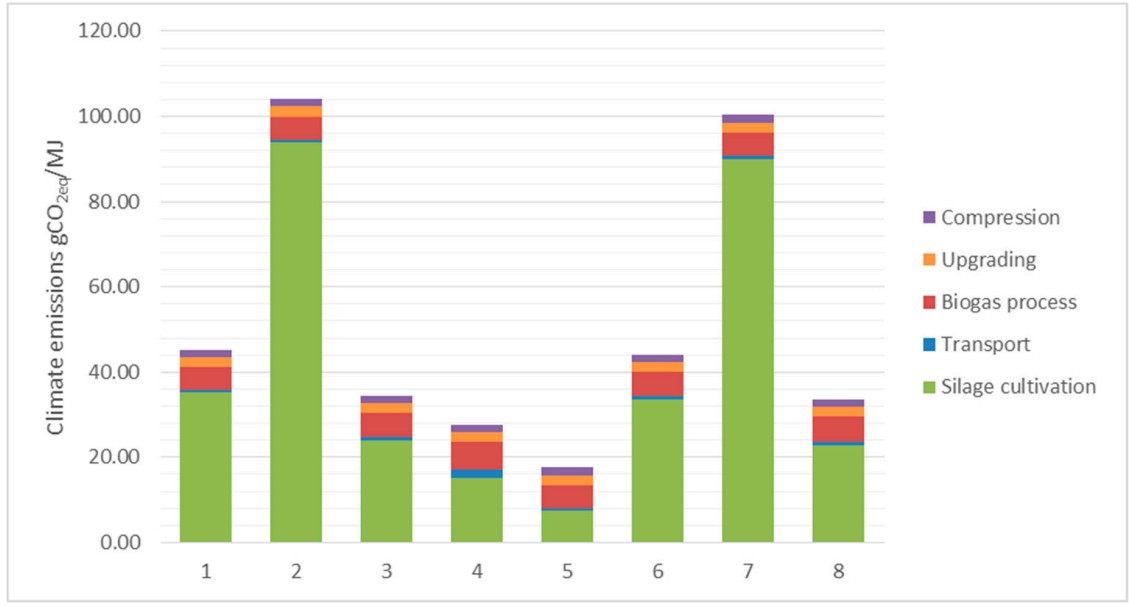

**Figure 2.** The share of GHG emissions from different stages of the processing chain (RED II manure bonus not included). 1. Grass silage (mineral soil). 2. Grass silage (organic soil). 3. Clover silage (mineral soil). 4. Grass silage (mineral soil) + 80% manure. 5. Grass silage from green manuring. 6. Grass silage (mineral soil) + 20% cattle manure. 7. Grass silage (organic soil) + 20% cattle manure. 8. Clover silage + 20% cattle manure.

**Table 5.** The share of GHG emissions from grass cultivation phase for different silage types.

| | Silage Type from Different Scenarios * | | 1 | 2 | 3 | 5 |
|---|---|---|---|---|---|---|
| | GHG Emissions | $gCO_{2e}kv/MJ$ | 28.2 | 75.3 | 20.1 | 6.1 |
| Share of emissions | Soil $N_2O$ emissions from fertiliser use | % | 54 | 15 | 50 | 0 |
| | Soil $N_2O$ emissions from decomposition of organic matter | % | 0 | 71 | 0 | 0 |
| | Soil $CO_2$ emissions from liming | % | 2 | 1 | 3 | 0 |
| | Production of fuels and use of machinery | % | 12 | 4 | 18 | 77 |
| | Production of mineral fertilisers | % | 27 | 7 | 21 | 0 |
| | Production of other inputs | % | 5 | 2 | 8 | 23 |

* 1 grass silage (mineral soil), 2 grass silage (organic soil), 3 clover silage (mineral soil), 4 grass from green manuring.

The lowest total GHG emissions were with grass silage from green manuring (scenario 5, 20 g $CO_{2eq}$/MJ). This is because no fertilisation was used, and only the emissions related to harvesting were taken into account (Table 5). RED II does not contain clear instructions concerning green manuring and, thus, this assumption of including only emissions from the harvesting was made by the authors to highlight the relevance of utilisation of side-products. The final word on how emissions of green manuring should be considered is with national implementation of RED II.

Manure as a co-substrate (scenarios 4, 6–8) for grass and clover always lowered the total GHG emissions due to its waste status in RED II. With waste materials, the emissions were allocated to the main product of the process (e.g., milk or meat in cases of manure) and not to the side stream. The emission reduction/credit received as a result was assumed to decrease the emissions originating from digestate storage in comparison to raw manure storage. This difference was estimated through "a manure bonus", which was credited only for the final emissions of the product (Table 6). Due to this, the total emissions in the manure scenarios (4, 6–8) decreased. The highest reduction was achieved in scenario 4 (from 28 $gCO_{2eq}$/MJ to 2 $gCO_{2eq}$/MJ), which co-digested manure as 80% of the feed, the rest being grass silage from mineral soils (Table 6). Manure as a substrate or the main substrate in a biogas plant gained small or even negative GHG emissions in other LCA studies as well [10,11,25–27].

**Table 6.** Emission reductions of biogas production compared to fossil fuel (94 $gCO_{2eq}$/MJ). For the scenarios with co-digestion with manure, the results are presented with and without manure bonus. Emission reduction potential includes manure bonus when applicable.

| Scenario | Total Emissions before Manure Bonus ($gCO_2eq/MJ$) | Total Emissions after Manure Bonus ($gCO_2eq/MJ$) | Emission Reduction Potential (%) |
|---|---|---|---|
| 1. Grass silage (mineral soil) | 45 | | 52 |
| 2. Grass silage (organic soil) | 104 | | −11 |
| 3. Clover silage | 34 | | 64 |
| 4. Grass silage (mineral soil) 20% + 80% manure | 28 | 2 | 98 |
| 5. Grass silage from green manuring | 18 | | 81 |
| 6. Grass silage (mineral soil) 80% + manure 20% | 44 | 42 | 55 |
| 7. Grass silage (organic soil) 80% +manure 20% | 100 | 98 | −4 |
| 8. Clover silage 80% + manure 20% | 34 | 31 | 67 |

In all scenarios, the direct and the indirect $N_2O$ emissions from grass cultivation contributed the most to the total emissions (7.6–93.9 $gCO_{2eq}$/MJ; Figure 2), as was also presented by Siddiqui et al. [25]. The second largest share of emissions came from the AD process (5.4 – 6.6 $gCO_{2eq}$/MJ) due to the external sources of electricity and heat needed. When biogas upgrading and compression were included, the emission increased to 9.4–10.6 $gCO_{2eq}$/MJ. However, if the electricity consumed during biomethane production (AD, upgrading and compression) was renewable or nuclear and thus had zero emissions, the emissions from the electricity consumed would decrease to 0.3–0.7 $gCO_{2eq}$/MJ

(fuel from loading substrate to the reactor and heat). This would reduce the total emissions significantly (8–95 $gCO_{2eq}$/MJ); Table 7).

**Table 7.** Emission reductions of biogas production in comparison to fossil fuel (94 $gCO_{2eq}$/MJ) if all consumed electricity is with zero emissions (renewable or nuclear) (the best case scenarios of zero-emissions in electricity production). For the scenarios with co-digestion with manure, the results are presented with and without manure bonus. Emission reduction potential includes manure bonus when applicable.

| Scenario | Total Emissions before Manure Bonus ($gCO_2eq$/MJ) | Total Emissions after Manure Bonus ($gCO_2eq$/MJ) | Emission Reduction Potential (%) |
|---|---|---|---|
| 1. Grass silage (mineral soil) | **36** | | 62 |
| 2. Grass silage (organic soil) | **95** | | −1 |
| 3. Clover silage | **25** | | 73 |
| 4. Grass silage (mineral soil) 20% + 80% manure | 18 | **−8** | 109 |
| 5. Grass silage from green manuring | **8** | | 74 |
| 6. Grass silage (mineral soil) 80% + manure 20% | 35 | **33** | 65 |
| 7. Grass silage (organic soil) 80% +manure 20% | 91 | **89** | 5 |
| 8. Clover silage 80% + manure 20% | 24 | **21** | 78 |

The lowest share of emissions was from transportation silage from the farm to the biogas plant (0.3–1.9 $gCO_{2eq}$/MJ). According to Adams and McMagnus [24], the highest GHG emissions in addition to soil $N_2O$ are from imported electricity and methane losses during the process. Methane losses from biogas plants can vary remarkably, but reported losses in traditional continuously stirred tank reactors are usually less than 1–4% of the produced methane [28–30]. The largest losses are normally due to open digestate storage tanks or, e.g., operational problems or foil roofs. Additionally, leakages in gas conducting plant components can occur [25]. In this study, the methane losses were estimated as negligible, as it was assumed that, in all scenarios, only covered digestate storage tanks were used.

According to RED II [4], for AD plants starting in early 2021 or later, the emission reduction requirement for biogas used in transport will be 65%. Of the scenarios assessed in this study, only the AD plant with grass silage from green manuring (scenario 5) and AD plants co-digesting manure (scenarios 4 and 8) achieved the required emission reduction (Table 6). Despite the fact that grass silage from mineral soil (scenarios 1 and 6, 45 and 44 $gCO_{2eq}$/MJ, respectively) and clover silage (scenario 3, 34 $gCO_{2eq}$/MJ) failed to fulfil RED II target, the biogas produced from these had lower emissions than fossil fuels (94 $gCO_{2eq}$/MJ; Table 6). The emission reduction of clover silage is very close to the limit value of RED II. It is thus noteworthy that any changes from the currently used average values for cultivation practices resulting in the same or higher yield with lower N-fertilising could also achieve the emission reduction. It can also be achieved by using manure as a co-substrate. For clover silage, a 20% manure share (scenario 8) is sufficient to reach the emission reduction, while with grass silage, the share of manure has to be larger. When grass from organic soil is digested, the total emissions (due to cultivation) are clearly higher than the fossil reference value. However, if the share of grass from organic soil is less than 20% of the total feed, emission reduction is reached (data not shown). In addition, if used electricity was produced from renewable sources, the total emissions from external energy use would decrease to 8–95 $gCO_{2eq}$/MJ and even to negative emissions (-8 $gCO_{2eq}$/MJ) when manure bonuses are included (scenario 4, Table 7). Due renewable electricity consumption, the AD plant with clover silage (scenario 2), grass silage from green manuring (scenario 5), and almost all of the AD plants co-digesting manure (scenarios 4, 6, and 8) achieved the required emission reduction.

$CO_2$ fluxes from the loss or gain of soil carbon are not included in the RED II method or this study. It should still be remembered that they may have some significance for the overall sustainability of renewable energy production. Greenhouse gas emission estimation methods capable of taking soil $CO_2$ fluxes into account are still being developed, and their application in LCA studies or RED II type

of initiatives is restricted by the high uncertainties related. Perennial grasses are known to be beneficial for carbon stocks in mineral soils in Finland [31,32] and in a lager view in Europe [33]. Thus, a growing trend of biomethane production based on grass leys could benefit the $CO_2$ balance of croplands by favouring diverse crop rotations with perennial species. Accordingly, grass as an energy crop could have benefits if it replaced annual cropping on organic soils. For example, the emission factor for $CO_2$ in organic soils diminishes by 8 t $CO_2$/ha/y when an annual crop is replaced by a perennial crop [34]. However, the effects of land clearance on carbon stock changes would be larger than these management-related impacts; thus, it is crucial to avoid carbon losses related to expansion of cultivated area. The RED II addresses this by restricting indirect land use changes resulting from the use of food or feed crops in bioenergy production.

The energy potential from agricultural plant-based side-streams, including unutilised grass and straw, is projected to vary from $1.2 \times 10^3$ to $2.3 \times 10^3$ PJ/year in the year 2030 in the EU28 [35], making these materials an attractive option to cover part of the renewable energy goals. Overall, biomethane can contribute to GHG emission reduction when sufficient attention is paid to the undesired emissions, e.g., from methane losses and dinitrogen oxide emissions from cultivation [36,37]. Additional benefits and emission reductions may be achieved, e.g., with efficient utilisation of the digestate, but its management and use are not included into the RED II calculation method. Recycling the nutrients back to the cultivation phase is needed to replace mineral fertilisers and the emissions related to them, but to account for the returned organic matter and its long-term impact on soil carbon requires more studies [24,37].

## 4. Conclusions

The results show that the origin of substrate affects the sustainability of biomethane production significantly. When using only grass silage as a substrate when the grass is cultivated exclusively for energy purposes, the emission reduction targets set in RED II are not easy to achieve. The reduction targets can be achieved if the grass is cultivated due to an improved crop rotation (e.g., for green manuring, maintaining good soil structure or increasing soil carbon stock) and only the emissions from harvesting are included. With the other grass types, co-digestion of grass with manure and/or using energy sources with zero emissions for the processes can be applied to assist in reaching the emissions targets. Additionally, cultivation of clover reduces the need for N fertilization and thus the emissions compared to cultivation of grass only, especially if soil $CO_2$ fluxes are taken into account. The information on GHG emissions of different substrates and their emission reduction rates is critical for guiding the inevitably expanding production of biofuels to a track of maximal sustainability.

**Author Contributions:** Conceptualization, S.R., K.R., P.V., H.H., E.T. and S.L.; methodology, K.T., K.J., P.V. and E.T.; formal analysis, K.T. and K.J.; writing—original draft preparation, S.R., K.T., K.J. and K.R.; writing—review and editing, P.V., H.H., E.T. and S.L.; supervision, S.R.; project administration, S.R.; funding acquisition, S.R. All authors have read and agreed to the published version of the manuscript.

**Funding:** This work was funded by Ministry of Agriculture and Forestry of Finland.

**Conflicts of Interest:** The authors declare no conflict of interest.

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
