# Peer review of "Sustainability of Vehicle Fuel Biomethane Produced from Grass Silage in Finland"

_sustainability, doi:10.3390/su12103994_

Round 1

Reviewer 1 Report

  1. please add taxonomical / latin names of grass and clover used. 
  2. Line 192-193 - please elaborate which emissions, is it the combustion of wood chips for energy?
  3. it would have been interesting to see one of the scenarios studied in reference to a certain geographical area in Finland. that gives an idea about applicability of scenarios such as co-digestion of grass from organic soil or manure. how much grass, clover, organic grass or manure is available in certain specific area?
  4. minor change - line 33 CO2 to CO2 

Author Response

1. taxonomical / latin names of grass and clover used are added to paragraph 2.1

2. The sentence "combustion of wood chips for energy" is added to text

3. The aim of the study was to show the possibility to use grass as raw material for biogas production and how different circumstances as soil type or co-digestion with manure has an effect on CO2 emissions. The certain case could have been calculated but we think it has only interest in Finland while the more general comparison is interesting in other countries as well, at least in EU. In Finland we also have an open service called Biomass Atlas that collects location data about biomass under a single user interface. This service can be used by anyone, who is interested to see, how much biomass (or grass) is available in certain areas.

4. Correction is made to the text

Reviewer 2 Report

The article is interesting and falls within the scope of this journal. I think it should be accepted as it is.

Author Response

Some small changes/additions according to other reviewer have been made. 

Reviewer 3 Report

Comments to the manuscript Sustainability-801425: Sustainability of Vehicle Fuel Biomethane Produced from Grass Silage in Finland

The manuscript impressively shows the relevance of sustainable grass cultivation on the GHG balance of biomethane, which seems to be an aspect of low relevance in the last years of biomethane and biogas production and corresponding funding. Hence, the study is a valuable issue to show the relevance of by-products and sustainable cultivation vs. exclusive cultivation of bioproducts, which are subsequently digested to form fuels. This topic clearly fits to the goals of the journal and is worthy to be published. However, a couple of minor aspects should be taken into consideration before publication:

  • Line 49: The advantage of the use of manure for biomethane production in case of reduced emissions of nitrate in context of the EU nitrate directive may be added.
  • Line 75: GHG emissions listed in RED II are mentioned. But it should be also mentioned in case of ‘direct emissions’ that a couple of Non-GHG emissions are also emitted and of high relevance due to legislative restrictions like formaldehyde or odor as sum parameter. The reference https://doi.org/10.1016/j.jclepro.2019.04.258 as example is dealing with these aspects and should be added to the manuscript.
  • As mentioned by the authors a real GHG reduction is not achieved by the grass if the grass is cultivated exclusively for matters of biogas production and GHG balancing is even negative. Dealing with these aspects the literature sources https://doi.org/10.1016/j.scitotenv.2014.02.038 and https://doi.org/10.1007/s10098-012-0568-0 should also be added to the manuscript.
  • Line 78: fulfil à fulfill
  • Line 101: …and use were included as they are cultivation, transportation….
  • Line 142: Please explain the abbreviation AIV
  • Table 3: The units of the N2O emissions are not clear. Is it kg N/ha again, or kg N2O/ha or kg CO2eq/ha? It seems to be kg kg CO2eq/ha due to the correction factor of 44. However, the higher GHG factor should be taken somehow into account. Nevertheless, these values seem to be very very low, i.e. 1 kg of Ammonia-N causes 15 g N2O, which is to low in case of manures as fertilizers. Please have a look on additional references improving this formula.
  • Figure 2: It might be interesting to add a reference scenario if possible, i.e. instead of biomethane production the GHG emissions of a typical scenario for production of conventional vehicle fuels. A value of 94 g CO2,eq/MJ is listed in Line 254. Might be very interesting to have a comparable differentiation to mining, transport, production in the distillation process…
  • Table 5: As a really small aspect, the sum of emissions in scenario 2 is 99% instead of 100%.
  • Line 250: …manuring (scenario 5) and AD plants co-digesting manure (scenarios 4 and 8)…
  • Table 6 and 7 headers: The reference value of 94 g CO2,eq/MJ should be added. It should be further mentioned that table 7 shows a best-case-scenario of zero emissions in electricity production.
  • Line 291: Please clarify the energy potential of unused grass and straw. Is this value for Finland or for the EU?

Author Response

Line 49: Yes, we acknowledge that the digestion of manure in a biogas plant transforms part of the organic nitrogen into  mineral form (NH4-N) and, thus, decreases the leaching potential as nitrate. NH4-N is more rapidly utilized in soils compared to the organic N, which is more slowly degrading and causes leaching of nitrate. However, the NH4-N in the digestate is easily volatilized as NH3, which could increase N emissions to air, if the digestate handling chain (storage, spreading) is not managed well. The use of manure as fertilizer is steered by the Nitrates Directive. In Finland, the whole country is considered as nitrate vulnerable zone, and the manure fertilization limit of 170 kgN/ha is implemented, which applies to both manure and digestates produced from manure.

The addition about NH4-N is now added to the text

Line 75: As the aim of the study was to calculate the emissions according to REDII, the non-GHG emissions from combustions is not included and no reference added to that. But other references about GHG balance is added to text.

Line 78: corrected

Line 101: corrected

Line 142: AIV is not abbreviation but the name of the preservative. But clarification is added to text 

Table 3: The units are added to table, as well as the correction to the last formula. The formulas are from IPCC rules, which are needed to be used in REDII calculation, this is why no changes were made to the formulas

Figure 2: Unfortunately, we had financing to make only comparison for biomethane although these other comparisons are interesting indeed, we have no resources to do them.

Table 5: the rounding of figures were reviewed and corrected

Line 250: corrected

Table 6 & 7: corrected

Line 291: corrected